# Differential Gene Expression Profiles between N-Terminal Domain and Ligand-Binding Domain Inhibitors of Androgen Receptor Reveal Ralaniten Induction of Metallothionein by a Mechanism Dependent on MTF1

**DOI:** 10.3390/cancers14020386

**Published:** 2022-01-13

**Authors:** Jon K. Obst, Nasrin R. Mawji, Simon J. L. Teskey, Jun Wang, Marianne D. Sadar

**Affiliations:** 1Department of Genome Sciences, British Columbia Cancer, 675 West 10th Avenue, Vancouver, BC V5Z 1L3, Canada; jobst@bcgsc.ca (J.K.O.); nmawji@bcgsc.ca (N.R.M.); simonteskey@gmail.com (S.J.L.T.); jeanwang@bcgsc.ca (J.W.); 2Department of Pathology and Laboratory Medicine, University of British Columbia, Vancouver, BC V6T 1Z7, Canada

**Keywords:** prostate cancer, metallothionein, androgen receptor, ralaniten, off-target, sintokamide, MTF1

## Abstract

**Simple Summary:**

Inhibition of the androgen receptor (AR) remains the mainstay treatment for prostate cancer. All current therapies involving AR inhibition either directly or indirectly target its ligand-binding domain (LBD). We have developed the first novel compounds which target the N-terminal domain, (NTD) a region which is essential for AR transcriptional activity. First-generation ralaniten (NCT02606123), and second-generation EPI-7386 (NCT04421222) remain the only AR-NTD inhibitors to progress to clinical trials. Here we aim to characterize differences between different classes of AR antagonists targeting the AR-LBD and the AR-NTD, as well as next generation AR-NTD inhibitors. An incidental finding was that ralaniten was uniquely associated with increased metallothionein expression which was independent of AR activity. Instead, expression of metallothionein genes was driven by MTF-1 indicating a potential off-target effect. Neither AR-LBD inhibitor enzalutamide nor second-generation AR-NTD inhibitor EPI-7170 had this effect. This work has important implications for the development of novel AR-NTD inhibitors.

**Abstract:**

Hormonal therapies for prostate cancer target the androgen receptor (AR) ligand-binding domain (LBD). Clinical development for inhibitors that bind to the N-terminal domain (NTD) of AR has yielded ralaniten and its analogues. Ralaniten acetate is well tolerated in patients at 3600 mgs/day. Clinical trials are ongoing with a second-generation analogue of ralaniten. Binding sites on different AR domains could result in differential effects on AR-regulated gene expression. Here, we provide the first comparison between AR-NTD inhibitors and AR-LBD inhibitors on androgen-regulated gene expression in prostate cancer cells using cDNA arrays, GSEA, and RT-PCR. LBD inhibitors and NTD inhibitors largely overlapped in the profile of androgen-induced genes that they each inhibited. However, androgen also represses gene expression by various mechanisms, many of which involve protein–protein interactions. De-repression of the transcriptome of androgen-repressed genes showed profound variance between these two classes of inhibitors. In addition, these studies revealed a unique and strong induction of expression of the metallothionein family of genes by ralaniten by a mechanism independent of AR and dependent on MTF1, thereby suggesting this may be an off-target. Due to the relatively high doses that may be encountered clinically with AR-NTD inhibitors, identification of off-targets may provide insight into potential adverse events, contraindications, or poor efficacy.

## 1. Introduction

Between 20 and 30% of patients with localized prostate cancer will experience recurrence and require systemic therapy. Androgen deprivation therapy (ADT) remains the first-line treatment; however, patients with metastatic disease will inevitably progress to castration-resistant prostate cancer (CRPC) despite initial response [1]. Following surgical or chemical castration, CRPC is clinically defined by an increase in the levels of serum prostate-specific antigen (PSA) which is an androgen receptor (AR)-regulated gene. These rises in serum PSA imply that the disease is still driven by AR transcriptional activity [2,3,4]. Second-generation hormonal therapies such as enzalutamide or abiraterone both directly or indirectly inhibit AR transcriptional activity by targeting the AR C-terminal ligand-binding domain (LBD) [5,6]. While initially effective, resistance to these therapeutics ultimately arises generally within 12–24 months [2,7]. Multiple mechanisms of resistance exist that continue to exploit the AR signaling pathway. These mechanisms include: amplification of the AR gene [8] and increased AR expression [9]; the emergence of AR-LBD mutations [10,11]; constitutively active AR splice variants (AR-Vs), which lack the LBD [12,13]; and de novo intratumoral androgen synthesis [7]. Unlike the AR-LBD, the N-terminal domain (NTD) is essential for transcriptional activities of both full-length AR and AR-Vs [14,15]. Preclinical studies demonstrating the utility of the first small-molecule inhibitors (ralaniten/EPI-002 and sintokamide/SINT1) designed to specifically target the AR-NTD have been described [16,17]. These inhibitors have shown success in the context of AR-Vs, AR gain-of-function mutations [17,18], amplified AR, altered AR NTD polyglutamine tract length, elevated levels of AR coactivators, and elevated levels of androgen [18,19]. In following, the prodrug of ralaniten (ralaniten acetate or EPI-506) and the second-generation EPI-7386 remain the first and only small-molecule inhibitors to have entered into Phase I clinical trials (Ralaniten, NCT02606123; EPI-7386, NCT04421222). Ralaniten acetate was well tolerated in heavily pretreated CRPC patients and imparted PSA responses and stable disease in some patients; however, it failed to advance to Phase II due to a poor pharmacokinetic profile. This proof of concept for the chemical structure and mechanism of action resulted in the development of more potent analogues with improved druglike qualities and metabolic stability. Thus, small molecules targeting the AR-NTD have a therapeutic niche, especially in the post-enzalutamide or -abiraterone setting [18,19,20]. Research and development of alternative compounds with improved potency, as well as combination therapies with EPI compounds, are ongoing [21,22,23,24], and a clinical trial investigating the combination of EPI-7386 with enzalutamide has begun (NCT05075577).

AR functions as a master regulator of thousands of genes and directly influences a myriad of cellular responses. Compounds that target the AR-NTD versus the AR-LBD have a markedly different mechanism of action, and potentially disrupt different downstream cellular pathways. Indeed, differences between ralaniten and SINT1 have been shown despite both targeting the AF-1 region within the AR-NTD [17]. Here, we begin to describe the molecular profile of ralaniten in an effort to better understand how AR-NTD inhibitors disrupt the molecular pathways under the control of the AR, and how this differs from existing AR-LBD inhibitors. In doing so, we aimed to discern whether the effects of ralaniten were specific to that compound, or were inherent to AR-NTD inhibitors as a whole. We found that both AR-LBD inhibitors and AR-NTD inhibitors were effective at preventing the induction of genes upregulated by the AR in response to androgen. Interestingly, differences between the two classes of inhibitors were much more apparent with respect to the ability to de-repress expression of androgen-repressed genes. Importantly, these studies led to the discovery of the unique induction of expression of metallothionein genes with ralaniten treatment. This unexpected result was not replicated with other AR-NTD inhibitors, and appeared to involve a mechanism that was independent of AR.

Metallothioneins are small (~6 kDa) cysteine-rich proteins that are capable of binding numerous essential as well as toxic metal ions; however, they preferentially form complexes with zinc [25,26]. Metallothioneins are ubiquitously expressed and play important roles in metal ion homeostasis and detoxification, as well as in protecting the cell from oxidative stress [27]. Metal response elements (MREs) have been defined in the proximal promoter elements of metallothionein members of the MT1 and MT2 subfamilies. Transcription of these genes is markedly enhanced following exposure to zinc and cadmium, primarily through the action of metal regulatory transcription factor 1 (MTF-1) binding to the MRE [28]. Since metallothioneins play a pivotal role in carcinogenesis and drug resistance [29], identifying the mechanism by which ralaniten increased their expression was of substantial interest.

## 2. Results and Discussion

### 2.1. Ralaniten Has a Unique Molecular Signature

To better understand the mechanism of action of ralaniten, and to identify differences between LBD antagonists, we performed a whole transcriptome microarray on cDNA isolated from LNCaP cells (Figure 1A), a model for which much of the preclinical work in developing ralaniten was performed, as well as decades of studies on androgen-regulated genes [16,19,30,31,32]. Cells were treated with enzalutamide (5 μM), bicalutamide (10 μM), ralaniten (35 μM), or DMSO vehicle and stimulated with 1nM R1881. We began by defining an androgen-responsive gene set. Protein-coding genes that were either induced (*n* = 456) or repressed (*n* = 209) ≥2-fold following stimulation with synthetic androgen R1881 were identified. Of these genes, the ability of each inhibitor to attenuate expression in the induced gene set, or de-repress expression in the repressed gene set by ≥2-fold was determined (Figure 1B,C). There was a large amount of overlap between all three inhibitors for attenuating the expression of androgen-induced genes. Closer inspection of the genes most induced by androgen revealed that all three compounds were capable of significantly reducing their expression (Figure 1D). Conversely, there was a much lower degree of commonality when we examined androgen-repressed genes. In this case, while bicalutamide and enzalutamide appeared to behave similarly with respect to specific genes each was able to de-repress, ralaniten had a unique signature. Furthermore, ralaniten had little effect on de-repressing the expression of the genes that were most susceptible to repression by androgen (Figure 1E). Both bicalutamide and enzalutamide were substantially superior in this regard compared to ralaniten. These data revealed for the first time the differences between AR-LBD inhibitors and AR-NTD inhibitors in gene expression. The differential impact between these two classes of inhibitors was most prominent with respect to disrupting the AR to de-repress androgen-repressed gene expression, which supported the role of specific AR domains in the multiple and complex mechanisms known to mediate transcriptional repression (for a review, see Grosse et al. 2012 [33]).

To determine what changes in gene expression were uniquely and significantly elevated in ralaniten-treated samples, hierarchical clustering was employed (Figure 1A). Unexpectedly, 5 of the top 11 genes that were positively associated with ralaniten-treatment were members of the metallothionein family (MT1F, MT2A, MT1G, MT1H, MT1X). Relative expression ranged from 6.52- to 18.40-fold over DMSO vehicle control (Log_2_FC = 2.711–4.125; Figure 1F), and notably, this effect did not occur with bicalutamide or enzalutamide. A subsequent GSEA was performed on the array data (Figure 1G–J) to obtain deeper insight into differences between the classes of AR inhibitors. As expected, the top gene set enriched in samples treated with androgen compared to DMSO vehicle control involved the androgen response (NELSON_RESPONSE_TO_ANDROGEN_UP, FDR = 0.000), and also as expected, all inhibitors had an overall negative expression pattern with genes in this set (Figure 1G). However, as seen in the earlier analysis (Figure 1C,E), ralaniten was unable to de-repress the expression of androgen-repressed genes (NELSON_RESPONSE_TO_ANDROGEN_DN, FDR = 0.000, Figure 1H). Furthermore, ralaniten was strongly associated with the increased expression of *MT1* and *MT2* genes (REACTOME_RESPONSE_TO_METAL_IONS, FDR = 0.000, Figure 1I; and REACTOME_METALLOTHIONEINS_BIND_METALS, FDR = 0.001, Figure 1J). Collectively, these data revealed that ralaniten: (1) was relatively poor at de-repressing the expression of androgen-repressed genes compared to bicalutamide and enzalutamide; and (2) uniquely induced the expression of genes within the metallothionein family.

### 2.2. Induction of Metallothioneins Is Specific to Ralaniten

To determine if the induction of metallothionein isoforms was specific to ralaniten or common to inhibitors of the AR-NTD, we employed a second-generation ralaniten analogue (EPI-7170) and two sintokamide analogues (SINT1 and LPY-26), which are structurally and mechanistically unique (Figure 2A,B). Sintokamide A (SINT1/compound 1; Figure 2B) is a natural compound that binds the AR AF1 region N-terminal to the binding site of ralaniten [17]. Functionally, SINT1 does not block STAT3 interaction with the AR-NTD, and consequently, it fails to block IL-6 transactivation of the AR-NTD, making it unique from ralaniten [16]. LPY-26 (compound 2) is a synthetic form of SINT1, and is presumed to bind in the same region as SINT1. EPI-7170 (compound 3) is a more potent analogue of ralaniten (compound 4) [22]. BADGE-2H_2_O (compound 5) has a high degree of structural similarity to ralaniten, yet does not bind AR and has no antagonistic effect against the AR [16], and was included as a control. Enzalutamide (compound 6) has no structural similarity to the NTD inhibitors, and targets the AR-LBD.

We first compared the AR-NTD inhibitors that bind to unique regions in the NTD to determine if the induction of expression of metallothionein genes was specific to ralaniten or common to NTD inhibitors. LNCaP human prostate cancer cells were transfected with luciferase reporter constructs fused to the promoter regions of metallothionein isoforms (MT1F, MT2, MT1G) found to be upregulated in the array data, and treated with 35 μM of each compound. Only ralaniten induced these metallothionein reporters (Figure 2C), suggesting that this effect was unique to ralaniten and not common to all NTD inhibitors. Consistent with the trends generated for the metallothionein reporter gene constructs, ralaniten also uniquely induced expression of endogenous metallothionein genes as determined by employing RT-PCR to measure levels of mRNA transcripts. Ralaniten significantly increased levels of expression of MT1F, MT1G, MT1X, and MT2A, whereas the more potent analogue EPI-7170 and the AR-LBD inhibitor enzalutamide did not (Figure 2D).

To confirm if induction of metallothionein genes by ralaniten occurred in vivo, castrated mice bearing LNCaP xenografts were randomized into three treatment groups: ralaniten (233 mg/kg), EPI-7170 (56.6 mg/kg), or vehicle control (CMC). Consistent with previous reports [21,22], both ralaniten and EPI-7170 were highly effective at preventing growth of these CRPC xenografts (Figure 2E), and had no significant impact on the body weights of animals (Figure 2F). Analyses of levels of expression of metallothionein genes from harvested xenografts were consistent with the in vitro data. Levels of expression of MT1F, MT1X, and MT2A were all significantly increased in xenografts harvested from animals dosed with ralaniten (Figure 2G). Interesting, EPI-7170 did increase expression of *MT1F* mRNA slightly but significantly compared to the vehicle control (CMC), but had no significant effect on levels of MT1X, MT1G, or MT2A.

### 2.3. Dose-Dependent Induction of Metallothionein by Ralaniten

To determine whether the induction of metallothionein expression was dependent upon concentration of ralaniten, MT1F and MT2 reporter gene constructs were transiently transfected into LNCaP cells. A dose-dependent response was observed, with significant induction occurring at 25 μM concentrations of ralaniten for both reporters (Figure 3A). Our next aim was to discover whether the induction of metallothionein expression by ralaniten was restricted to LNCaP cells, or could be replicated in additional models. To this end, LNCaP95 (LN95) cells were utilized. Despite expressing functional full-length AR (AR-FL), these cells are reliant upon the splice variant AR-V7 to drive proliferation. Therefore, the dose dependence of ralaniten on induction of endogenous expression of metallothionein genes was also measured in LNCaP and LN95 cells. Ralaniten strongly induced expression of metallothionein isoforms in a dose-dependent response in both of these cell lines, demonstrating that the induction of metallothionein expression by ralaniten was not specific to LNCaP cells. This effect was not measured with any concentration of enzalutamide (Figure 3B). Interestingly, EPI-7170 had a similar effect, but only at the highest concentration tested (20 μM) in LN95 cells (Figure 3C). EPI-7170 at 20 μM was cytotoxic in LNCaP cells.

### 2.4. Induction of Metallothionein Isoforms by Ralaniten Is not Dependent upon the AR

Expression of metallothionein genes are regulated through a number of response elements within its regulatory regions. These include binding sites for glucocorticoid receptor (GR), NRF2, and MTF-1 [28,34,35], which were within the metallothionein reporter gene constructs employed here (for a review, see Coyle et al. 2002 [35]). Since AR and GR can overlap in binding some shared-response elements on genes [36,37], we next determined if AR played a role in the induction of metallothionein genes by ralaniten. To do this, we first determined if expression of AR was required for ralaniten induction of expression of metallothionein genes. Metallothionein gene expression was assayed in several additional prostate cancer cell lines that lack AR expression (DU145) or express very low levels of nonfunctional AR (PC3). Both the LNCaP and LN95 cell lines were included, as they each had demonstrated increased metallothionein expression in response to ralaniten treatment previously (Figure 3B). Overall, LN95 cells behaved similarly to LNCaP cells that only express full-length AR, with ralaniten significantly inducing metallothionein expression (Appendix A). Conversely, neither DU145 nor PC3 cells demonstrated any statistically significant induction of expression of metallothionein following ralaniten treatment (Appendix A). As seen previously in LNCaP cells, both enzalutamide and EPI-7170 also failed to induce expression of metallothionein in any cell line. Collectively, these data implied that the ability of ralaniten to induce expression of metallothionein genes may be reliant upon functional full-length AR, or alternatively, may be cell-specific.

To more definitively address the requirement for AR in the mechanism of induction of metallothionein isoforms by ralaniten, levels of AR were knocked down in LNCaP and LN95 cells, and subsequently treated with DMSO vehicle (in the presence and absence of androgen), enzalutamide (5 μM), ralaniten (35 μM), or EPI-7170 (5 μM), all in the absence of androgen stimulation. While AR protein levels were robustly silenced after 24 h in LNCaP cells (Figure 4A), the effect was lessened in LN95 cells (Figure 4B and Appendix A for uncut gels). Nonetheless, treatment with AR siRNA alone was sufficient to reduce expression of canonical AR-regulated genes (KLK3/PSA and FKBP5) in the presence of synthetic androgen R1881 in both cell lines, supporting a reduction in AR-FL (Figure 4C,D). Ralaniten retained the ability to induce expression of metallothionein, despite the loss of AR levels and subsequent activity in both cell lines (Figure 4E,F). There was little to no decrease in mRNA expression levels of any metallothionein isoform induced with ralaniten with AR knockdown. Collectively, these data supported that ralaniten-induced expression of metallothionein was independent of AR, and suggested that this induction may be an off-target of ralaniten.

### 2.5. Induction of Metallothionein Isoforms by Ralaniten Is Not Associated with NRF2 Expression or Increased Oxidative Stress

The genes for metallothionein contain antioxidant response elements (ARE) to which NRF2 binds to increase levels of expression in response to oxidative stress [34,38,39]. Both ralaniten and EPI-7170 increased levels of NRF2 protein in LNCaP and LN95 cells (Appendix A, and Appendix A for uncut gels). In agreement with increased levels of NRF2 protein, ralaniten and EPI-7170 also induced levels of canonical NRF2-regulated genes, NQO1 and SLC11A7, which could be blocked by knockdown of NRF2 (Appendix A). Importantly, knockdown of NRF2 did not reduce ralaniten-induced expression of metallothionein in either LNCaP or LN95 cells (Appendix A). These data indicate that ralaniten induction of metallothionein was driven by a mechanism independent of NRF2.

Oxidative stress has been shown to lead to zinc disassociation from metallothionein proteins, which could also lead to an increase in free zinc ions to activate MTF-1 independently of NRF2 [28]. To test this, the fluorescent probes MitoSOX and 2′,7′-dichlorofluorescin diacetate (DCFDA) were used to detect production of intracellular ROS following treatment with ralaniten, EPI-7170, enzalutamide, and the positive control, *tert*-butyl hydroperoxide (tBHP). Neither enzalutamide nor the AR-NTD inhibitors mediated a positive signal for mitoSOX (Appendix A), whereas a modest yet significant increase in ROS was seen only for enzalutamide compared to DMSO control, and only in LNCaP cells (Appendix A). Neither ralaniten nor EPI-7170 mediated any increase in ROS generation (Appendix A).

### 2.6. Induction of Metallothionein Isoforms by Ralaniten Is Dependent upon MTF1

Transcription of metallothionein subfamilies MT1 and MT2 is markedly enhanced through MTF-1 binding to MREs within their proximal promoters [28]. The transcriptional activity of MTF-1 is regulated by nuclear localization and a metal-responsive transactivation domain [27,28,34]. To determine the role of MTF-1 in the mechanism of induction of metallothionein by ralaniten, levels of this transcription factor were measured in response to increasing concentrations of ralaniten and in the context of knockdown of MTF-1 in LNCaP (Figure 5A) and LN95 cells (Figure 5B). In LNCaP cells, both androgen and ralaniten slightly elevated levels of expression of MTF-1, although this was not significant (Figure 5A). Interestingly in LN95 cells, ralaniten at 35 μM approximately doubled the levels of expression of MTF-1, an effect that was more pronounced in the sample treated with control siRNA (Figure 5B). Regardless, knockdown of MTF-1 significantly blocked ralaniten induction of expression of metallothionein isoforms (MT1F, MT1X, MT2A) in both cell lines (Figure 5A,B). These data suggested that ralaniten-induced expression of metallothionein was dependent upon expression levels of MTF-1.

To distinguish the ability of ralaniten to inhibit AR signaling in the context of knockdown of MTF-1, levels of expression of androgen-regulated genes (KLK2, KLK3/PSA, and FKBP5) were measured. Targeted knockdown of MTF-1 was repeated in LNCaP and LN95 cells treated with increasing concentrations of ralaniten and stimulated with R1881. Ralaniten inhibited the levels of expression of androgen-regulated genes in a dose-dependent manner regardless of knockdown of MTF-1 (Figure 5C,D). However, knockdown of MTF-1 significantly reduced expression levels of KLK2 and KLK3 in androgen-induced LNCaP cells (Figure 5C). Interestingly, this was not observed in LNCaP cells for the expression of FKBP5 or for any of these three genes (KLK2, KLK3, FKBP5) in LN95 cells (Figure 5D). The ability of ralaniten to inhibit androgen-induced cellular proliferation in LNCaP (IC_50_ SCRM: 13.26 μM; MTF-1: 11.25, *p* = 0.405; Figure 5E), or androgen-independent proliferation of LN95 cells was not affected by knockdown of MTF1 (IC_50_ SCRM: 16.14 μM; MTF-1: 13.52, *p* = 0.141; Figure 5F).

## 3. Materials and Methods

### 3.1. Chemicals and Compounds

Metribolone (R1881) was purchased from AK Scientific (Mountainview, CA, USA); the antiandrogen bicalutamide was a kind gift from Mark Zarenda (AstraZeneca) and the antiandrogen enzalutamide was purchased from Omega Chem (Lévis, QC, Canada). Ralaniten was provided by Naeja-RGM (Edmonton, AB, Canada). SINT1 is a natural compound, and EPI-7170 and LPY-26 were synthesized by Raymond Andersen at UBC (Vancouver, BC, Canada). All other chemicals, including BADGE-2H_2_O, were purchased from Sigma Aldrich (St. Louis, MO, USA) unless stated otherwise.

### 3.2. Cell Culture

LNCaP cells were from Leland Chung (Cedars-Sinai Medical Center, Los Angeles, CA, USA) and maintained in phenol-red-free RPMI 1640 supplemented with 5% FBS (VWR). LN95 cells were provided by Stephen Plymate (University of Washington). LN95 cells were maintained in RPMI 1640 medium supplemented with 10% dextran-coated charcoal-stripped FBS. DU145 cells were from Victor Ling (British Colombia Cancer Agency, Integrative Oncology) in October 1998. Cells were maintained in DMEM with 10% FBS and supplemented with 2 mM L-glutamine and 1 mM of sodium pyruvate. PC3 cells were purchased from ATCC and maintained in DMEM with 5% FBS and supplemented with 2 mM L-glutamine and 1 mM of sodium pyruvate. LN95 cells were not authenticated in our laboratory, but were regularly tested to ensure that they were mycoplasma-free (VenorTMGeM Mycoplasma Detection kit, Sigma-Aldrich). LNCaP, PC3, and DU145 cells were authenticated by short tandem repeat analysis and tested to ensure that they were mycoplasma-free by DDC Medical. All cells used in the experiments were passaged in our laboratory for fewer than 3 months after resurrection.

### 3.3. Microarray and GSEA Analysis

Total RNA was extracted from LNCaP cells treated with ralaniten (35 µM), enzalutamide (5 µM), bicalutamide (10 μM), or DMSO vehicle and stimulated with either 1 nM R1881 or EtOH vehicle. RNA was reverse-transcribed, and cDNA was hybridized to the GeneChip Human Transcriptome Array 2.0 from Affymetrix. RT-PCR, cDNA hybridization, and chip reading was carried out at CDRD’s Target Validation Division at the University of British Columbia (Vancouver, BC, Canada; www.cdrd.ca (14 October 2014)). Analysis of raw signal output was done using GeneSpring software (version 13.1). Clustered data was generated by conducting a two-way ANOVA on data with the significance threshold set at 0.05. The Benjamini–Hochberg correction was applied to reduce the false discovery rate (FDR).

Data generated from the microarray analysis was used to identify differences between AR antagonists. GSEA version 7.0 software (http://software.broadinstitute.org/gsea/msigdb/index.jsp (5 December 2019) was used, and the difference in the expression levels between vehicle and drug treatment for each gene was analyzed based on the Molecular Signatures Database C2 sets (KEGG gene sets, c2.all.v6.2.symbols.gmt). The permutation number was set to 1000. Those enrichment gene sets revealed by GSEA as exhibiting a nominal *p* < 0.05 and FDR < 0.05 were considered to indicate a statistically significant difference. The default parameters were used in the GSEA software.

### 3.4. Reporter Assays

LNCaP cells were transfected with reporter plasmids pMT1F-luciferase, pMT1G-luciferase, or pMT2-luciferase (1 μg/well) in serum-free, phenol-red-free media using Lipofectin (Invitrogen, Waltham, MA, USA). After 16 h, cells were treated with SINT1 (35 μM), LPY-26 (35 μM), BADGE-2H_2_O (35 μM), ralaniten (35 μM), or DMSO vehicle. After 48 h of treatment, cells were lysed and analyzed for luciferase activity and normalized to protein expression. For dose-escalation experiments, LNCaP cells were transfected with the reporter plasmid pMT1F-luciferase, pMT1G-luciferase, or pMT2-luciferase in serum-free, phenol-red-free media using Lipofectin (Invitrogen). After 16 h, cells were treated with increasing concentrations of ralaniten or DMSO vehicle. After 48 h of treatment, cells were lysed and analyzed for luciferase activity and normalized to protein expression.

### 3.5. Xenografts and Animal Experiments

All experiments involving animals were conducted in compliance with, and with the approval of, the Animal Care Committee of the University of British Columbia (A18-0077). Male NOD/SCID mice at 6 to 8 weeks of age were subcutaneously injected with LNCaP cells (1 × 10^7^ cells/site) using Matrigel (Becton Dickinson). Mice were castrated once tumors reached ~100 mm^3^ and randomized into treatment groups to receive ralaniten (233 mg/kg), EPI-7170 (56.6 mg/kg) or vehicle control (5% DMSO/1% CMC/0.1% Tween 80) once daily by oral gavage. Treatment began one week following castration. Mouse body weight and tumor volume (defined as volume = length × width × height × 0.5236) were regularly recorded, and tumors were excised 24 h after the last treatment. To analyze tumor gene expression, tumors were flash frozen, and ~100 mg was added to 1 mL TRIzol reagent (Invitrogen) and homogenized using a tissue homogenizer (MP Biomedicals). RNA was extracted and reverse-transcribed as detailed below.

### 3.6. Gene-Expression and Dose-Escalation Experiments

LNCaP, LN95, DU145, and PC3 cells were plated on 6-well plates in respective full media for 24 h. Cells were serum starved for 24 h in serum-free RPMI 1640 prior to treatment. Cells were treated with ralaniten (0.5–35 μM), enzalutamide (0.5–20 μM), EPI-7170 (0.5–20 μM), or DMSO vehicle in the absence of androgen stimulation. After 24 h of treatment, cells were harvested in 1 mL TRIzol reagent (Invitrogen). Total RNA was extracted using the RNeasy Micro Kit (Qiagen, Hilden, Germany), cleaned using an amplification-grade DNase I Kit (MilliporeSigma, Burlington, MA 01083, USA) and reverse-transcribed using the High-Capacity RNA-to-cDNA Kit (Thermo Fisher Scientific, Waltham, MA, USA). Diluted cDNA and Platinum SYBR Green qPCR SuperMix-UDG with ROX (Invitrogen) were combined with gene-specific primers. Transcripts were measured by a qRT-PCR QuantStudio 6 Flex Real-Time PCR System (Applied Biosystems by Life Technology), and gene expression was normalized to the housekeeping gene SDHA. Gene-specific primer sequences are given in Appendix A.

### 3.7. siRNA Knockdown Experiments

Pooled siRNA against AR (L-003400-00-0005), MTF-1 (L-020078-00-0005), NRF2 (L-003755-00-0005), and nontargeting control (D-001810-10-05) were purchased from Dharmacon Research (Layfayette, CO, USA). Lipofectamine RNAiMAX (Invitrogen) was used to transfect 10 nM (AR, NRF2) or 15 nM (MTF-1) of siRNA into cells in Opti-MEM serum-free media (Thermo Fisher Scientific). For experiments examining protein or mRNA expression, cells were plated on 10 cm or 6-well plates, respectively, in complete media for 24 h prior to transfection. After 24 h, media were removed and replaced with Opti-MEM (Gibco) containing 10 nM (AR, NRF2) or 15 nM (MTF1) siRNA/transfection reagent complexes (Lipofectamine RNAiMAX Transfection Reagent, Invitrogen). Cells were pretreated with ralaniten (5–35 µM), EPI-7170 (5 µM), enzalutamide (5 µM), or DMSO vehicle for 1 h, followed by stimulation with 1 nM R1881 or EtOH. Cells were harvested in 1 mL TRIzol, and RNA was extracted and reverse-transcribed as detailed above.

Protein lysates were harvested 24 h after R1881/EtOH treatment in RIPA buffer and separated on a 10% SDS-PAGE gel and transferred to a PVDF membrane (Millipore LTD, Cork, IRL). AR was probed using an antibody obtained from Abcam (Ab198394). MTF-1 was probed using an antibody obtained from Invitrogen (PA5-55945). NRF2 was probed using an antibody obtained from Abcam (Ab137550). β-actin was used as a loading control, and membranes were probed using the mouse monoclonal anti-β-actin antibody (a5316 from Sigma).

For the proliferation assay, 5000 (LNCaP) or 7500 (LN95) cells were plated per well in 96-well plates in respective full media and incubated for 24 h to allow cells to attach. Media were removed and replaced with Opti-MEM containing 15 nM MTF1 siRNA or scrambled control for 24 h. Treatments were prepared by serial dilution, and cells were pretreated with DMSO or ralaniten (5–35 µM) for 1 h prior to stimulation with 0.1 nM R1881 (LNCaP) or 1.5% charcoal-stripped serum (LN95). After 72 h post-treatment, cells were fixed in 4% paraformaldehyde (Electron Microscopy Sciences) and incubated with 0.1% crystal violet solution (Sigma). Dye was solubilized using a 1% SDS solution, and absorbance was read using a VersaMax Microplate Reader (Molecular Devices) at 595 nm.

### 3.8. ROS Detection

Total cellular ROS in cells was detected using the DCFDA/H2DCFDA-Cellular ROS kit from Abcam (ab11351) according to the manufacturer’s instructions. LNCaP and LN95 cells plated on 96-well, black, clear-bottom plates were stained with DCFDA/H2DCFDA (20 μM) for 30 min at 37 °C. After washing, cells were treated with tBHP (75 μM), enzalutamide (5 μM), ralaniten (35 μM), EPI-7170 (5 μM), or DMSO vehicle for 4 h. Fluorescence was measured using the Infinite M1000 (Tecan) with excitation at 485 nm and emission at 535 nm.

Mitochondrial superoxide was detected using MitoSOX Red from Sigma (M36008). LN95 cells were seeded on 8-well chamber slides and treated with tBHP, enzalutamide, ralaniten, EPI-7170, or DMSO vehicle as above for 4 h at 37 °C. Cells were then washed and stained with MitoSOX Red (5 μM) diluted in HBSS/Ca/Mg buffer (Gibco, 14025-092) for 10 min at 37 °C. Cells were washed gently three times in warm buffer before counterstaining with Hoechst 33342 (Sigma, 62249). Slides were mounted and visualized using a fluorescence microscope (Axio Imager.M2, ZEISS) at Ex/Em 350/461 (Hoechst) and 510/580 (MitoSOX Red).

### 3.9. Statistical Analysis

A one- or two-way ANOVA statistical test was used to determine significance for all comparisons unless specifically stated otherwise (Graphpad Prism, version 7.0). The *p*-value corrections were applied for all multiple comparisons (Tukey, Sidak, or Dunnett, as appropriate), and *p* < 0.05 was considered statistically significant.

## 4. Conclusions

Despite initial response to targeted AR inhibition, most CRPC remains driven by continued AR transcriptional activity. Currently, there is no cure for CRPC, and patients will inevitably succumb to their disease. The existence of constitutively active AR-Vs, gain-of-function mutations within the AR-LBD, amplification of the AR gene, and intratumoral synthesis of androgens have been shown drive resistance to abiraterone and enzalutamide [11,13,14]. Additional evidence has implicated a genomic basis conferring an enhanced risk of prostate cancer. Germline mutations in a number of DNA-repair genes (especially BRCA2) have been associated with more aggressive disease and poor clinical outcomes [40]. Exploiting alterations in these genes allows their use as additional biomarkers to guide treatment decisions, best exemplified by the recent success of the Phase III clinical trial investigating the use of the PARP inhibitor olaparib compared to conventional AR-targeted therapies (NCT02987543) [41].

Furthermore, the discovery and development of small-molecule inhibitors that target the AR-NTD have significant potential to improve outcomes of CRPC patients who have progressed following treatment with existing AR-targeted therapies [16,18,19,42]. The prodrug of ralaniten (EPI-506) represents the first such compound to advance into clinical trials for heavily pretreated men with CRPC that had failed abiraterone and/or enzalutamide. This trial provided a proof of concept for the molecular scaffold of ralaniten, and showed signs of efficacy in spite of poor pharmacokinetics. These clinical studies have encouraged the clinical development of second-generation EPI compounds with improved potency and druglike qualities such as EPI-7386 (NCT04421222). Thus, ralaniten remains a useful tool compound to model and describe the mechanism of action of AR-NTD inhibitors. Here, we have begun to lay the groundwork for defining the molecular changes that occur following exposure to ralaniten. Specifically, we have shown that ralaniten: (1) was remarkably similar to bicalutamide and enzalutamide in its inhibition of expression of androgen-induced genes; (2) deviated substantially from AR-LBD inhibitors in de-repressing the expression of androgen-repressed genes, supporting the complexity of repression mechanisms that involve multiple mechanisms; and (3) uniquely induced the expression of metallothionein genes by a mechanism that was dependent on levels of MTF-1. Ralaniten induction of expression of the metallothionein genes was independent of the AR, thereby implying that this was an off-target and unique to ralaniten. Metallothioneins also play important roles in tumor growth, progression, and drug resistance, and may play a protective role due to being positively associated with malignancy and tumor grade in a number of human cancers [29,43]. Interestingly, this relationship seems to be at least partially reversed in prostate cancer, with expression of some metallothionein isoforms significantly downregulated or lost in advanced disease [44,45,46,47]. Here, we revealed that the efficacy of ralaniten as an inhibitor of AR-dependent growth was not adversely impacted by alterations in MTF1 and metallothionein.

## Figures and Tables

**Figure 1 cancers-14-00386-f001:**
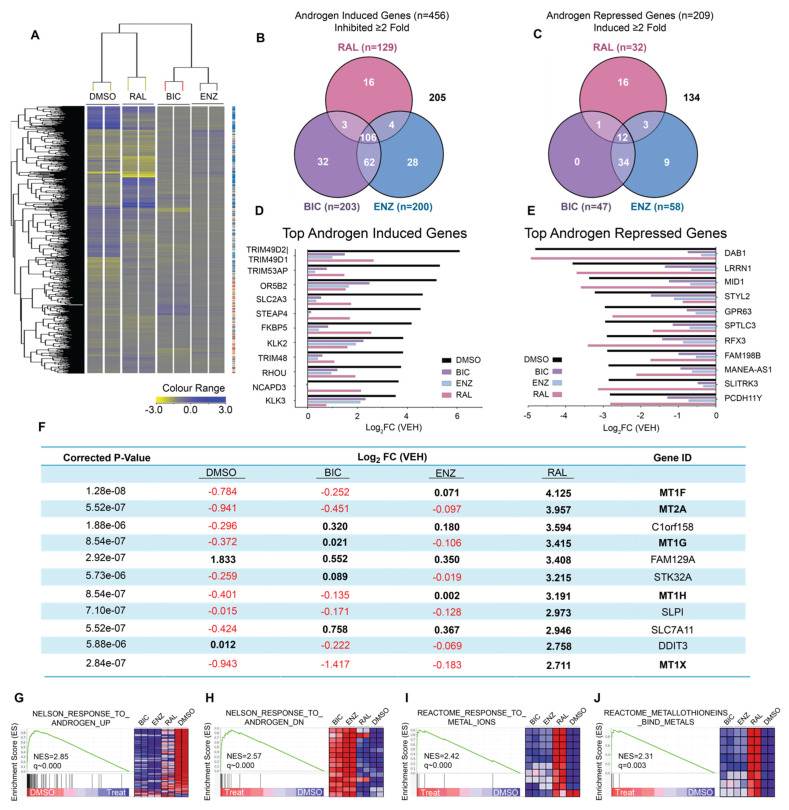
Differential gene expression profiles between AR-LBD inhibitors and an AR-NTD inhibitor. (**A**) Heatmap showing differentially regulated genes following treatment with bicalutamide (10 µM), enzalutamide (5 µM)**,** ralaniten (35 µM), or DMSO vehicle (R1881 only), and all stimulated with 1 nM R1881 for 24 h. Data were normalized to vehicle treated with EtOH (no R1881). Analysis of global gene expression identified androgen-regulated genes that were specifically induced (*n* = 456) (**B**) or repressed (*n* = 209) (**C**). The Venn diagrams show the numbers of androgen-regulated genes in these subsets whose expression decreased/increased by ≥2-fold in the presence of bicalutamide, enzalutamide, or ralaniten. Numbers outside Venn diagrams indicate genes that were not affected by any drug treatment. Top androgen-induced (**D**) and -repressed (**E**) genes ranked by fold-change normalized to DMSO/EtOH (no R1881). LNCaP cells were incubated with enzalutamide (ENZ, 5 μM), bicalutamide (BIC, 10 μM), ralaniten (Ral, 35 μM), or DMSO vehicle (with and without 1 nM R1881). All drug treatments were stimulated with 1 nM R1881 for 24 h before harvesting and isolation of RNA for analyses. (**F**) Table showing the top genes that positively correlated with ralaniten treatment. Normalized expression values which are reduced compared to vehicle control are indicated in red, expression values which are increased, are in bold. Five genes clustered within the metallothionein family showed significant enrichment, and were specifically associated with ralaniten treatment. (**G**) GSEA plot of top gene set that showed enrichment in DMSO control (R1881 only) compared against drug treatment. (**H**–**J**) GSEA plots of top gene sets that showed enrichment in drug-treated samples compared to DMSO control (R1881 only). Associated heatmaps show expression patterns of core enriched genes for each treatment. Ralaniten had a unique signature, while bicalutamide and enzalutamide had similar expression patterns. All data were normalized to DMSO control in the absence of R1881 (EtOH). DMSO means R1881 treatment only. Bicalutamide, enzalutamide, and ralaniten were all in the presence of R1881.

**Figure 2 cancers-14-00386-f002:**
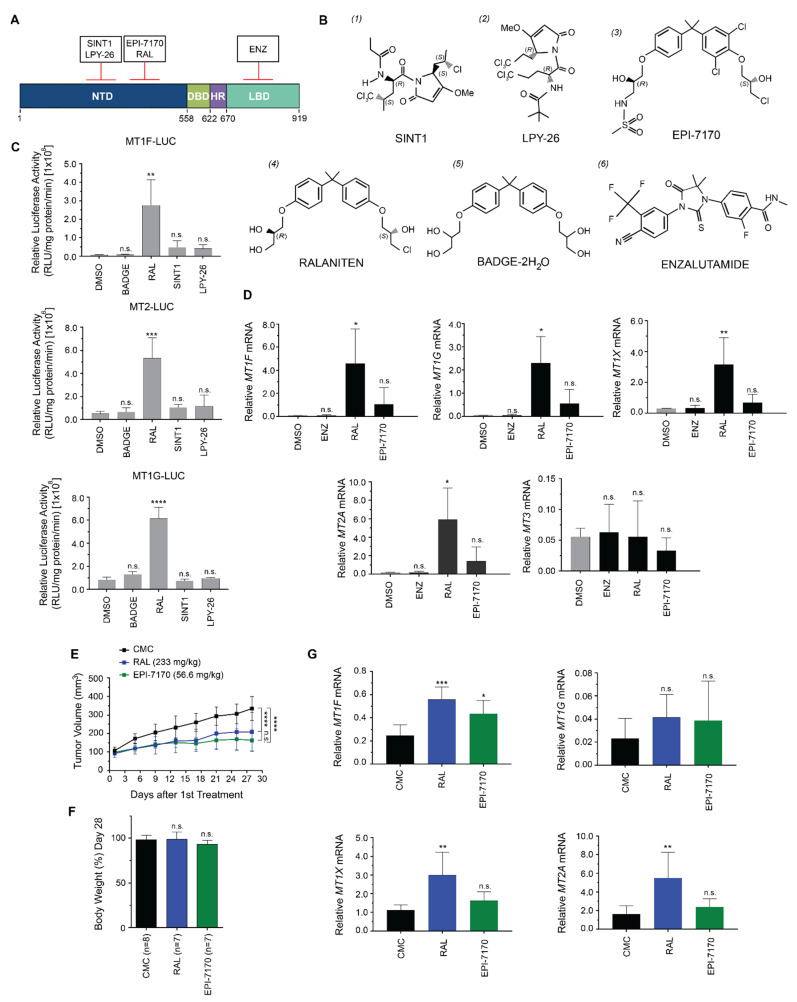
Ralaniten-induced expression of metallothionein in vitro and in vivo. (**A**) Illustration depicting the various compounds tested with respect to which region they bind on AR. (**B**) Chemical structure of each compound. BADGE-2H_2_O does not bind AR and has no inhibitory effect on AR transcriptional activity. (**C**) LNCaP cells transfected with the various MT1/2-luciferase reporters and treated with DMSO, BADGE-2H_2_O (35 μM), ralaniten (35 μM), SINT1 (35 μM), or LPY-26 (35 μM) for 24 h. Data presented as mean ± SD and analyzed by one-way ANOVA with Dunnett’s test applied post hoc (*n* = 3 independent experiments). (**D**) Transcript levels of MT1F, MT1G, MT1X, MT2A, and MT3 normalized to SDHA from LNCaP cells treated with enzalutamide (5 µM), ralaniten (35 µM), EPI-7170 (5 μM), or *v*/*v* DMSO. Data presented as mean ± SD and analyzed by two-way ANOVA with Sidak’s test applied post hoc (*n* = 3 independent experiments). (**E**) Ralaniten and EPI-7170 inhibited the growth of LNCaP CRPC xenografts in castrated hosts. One week after castration, when tumors were approximately 100 mm^3^, animals were dosed daily by gavage with CMC vehicle (3% DMSO, 1.5% Tween 80, 1% CMC), Ral (233 mg/kg body weight), or EPI-7170 (56.6 mg/kg body weight). Mean ± SD of *n* = 9 or 10 for each group. (**F**) Body-weight change over the course of the experiment. Data presented as mean ± SD and analyzed by two-way ANOVA with Dunnett’s test applied post hoc (*n* = 8 or 7 mice per group). (**G**) Real-time PCR for measurement of levels of MT1F, MT1G, MT1X, and MT2A transcripts that were normalized to levels of SDHA transcript using RNA harvested from LNCaP xenografts (*n* = 6, CMC; *n* = 4, RAL; *n* = 7, EPI-7170). * *p* < 0.05; ** *p* < 0.01; *** *p* < 0.001; **** *p* < 0.0001; n.s., not significant. ENZ, enzalutamide, RAL, ralaniten.

**Figure 3 cancers-14-00386-f003:**
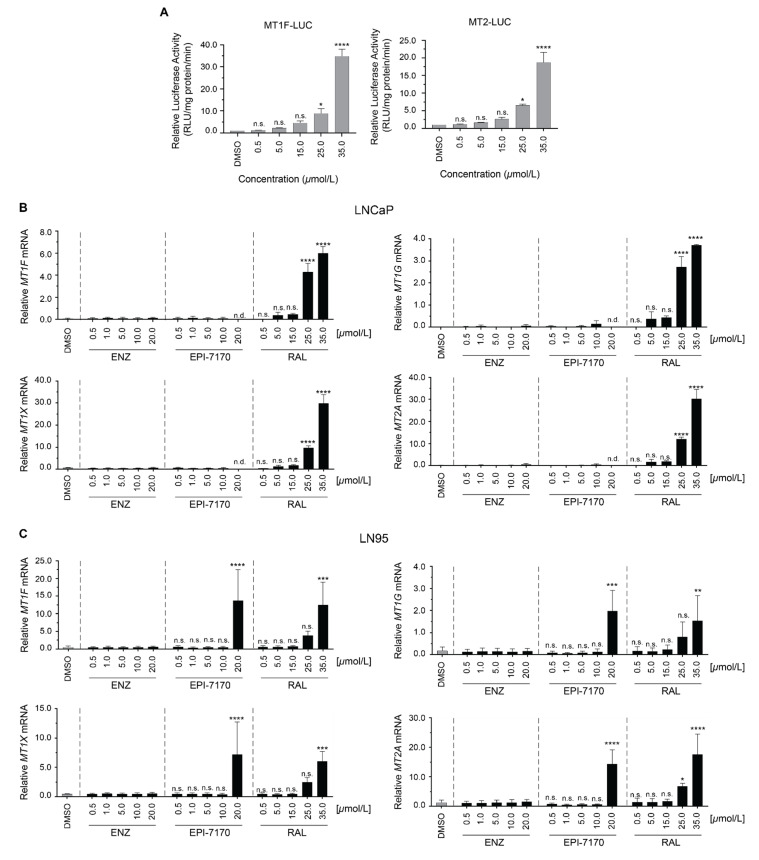
Induction of metallothionein gene expression by ralaniten is dependent on concentration. (**A**) LNCaP cells were transiently transfected with MT1-luciferase or MT-2-luciferase reporters. Cells were subsequently treated with DMSO or increasing concentrations of ralaniten (0.5–35 μM) for 24 h. (**B**) LNCaP and (**C**) LN95 transcript levels of MT1F, MT1G, MT1X, and MT2A normalized to levels of SDHA transcript from cells treated with DMSO, or increasing concentrations of enzalutamide, EPI-7170, and ralaniten for 24 h. Data presented as mean ± SD and analyzed by two-way ANOVA with Dunnet’s test applied post hoc (*n* = 3 independent experiments). EPI-7170 was toxic at 20 μM in LNCaP cells. * *p* < 0.05; ** *p* < 0.01; *** *p* < 0.001; **** *p* < 0.0001; n.s., not significant. ENZ, enzalutamide; RAL, ralaniten; n.d., not detected.

**Figure 4 cancers-14-00386-f004:**
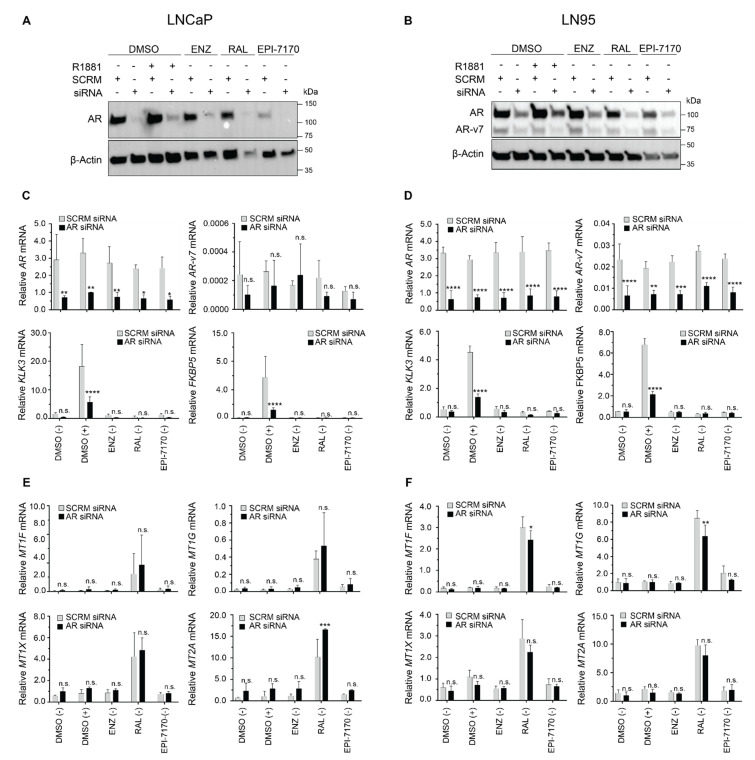
Ralaniten-induced expression of metallothionein is independent of levels of AR. (**A**) Western blot analyses of levels of AR proteins in LNCaP and (**B**) LN95 cells transfected with scramble (SCRM, control) or AR siRNA (10 nM) for 24 h and treated with DMSO, enzalutamide (5 μM), ralaniten (35 μM), or EPI-7170 (5 μM) in the presence or absence of 1 nM R1881. The uncropped immunoblot images can be found in Appendix A. (**C**) Transcript levels of AR, AR-V7, KLK3, and FKBP5 were normalized to transcript levels of *SDHA* from LNCaP and (**D**) LN95 cells treated with DMSO, enzalutamide (5 μM), ralaniten (35 μM), or EPI-7170 (5 μM) in the presence (+) or absence (−) of 1 nM R1881. (**E**) Transcript levels of *MT1F*, *MT1G*, *MT1X,* and *MT2A* were normalized to transcript levels of *SDHA* from LNCaP and (**F**) LN95 cells treated with DMSO, enzalutamide (5 μM), ralaniten (35 μM), or EPI-7170 (5 μM) in the presence (+) or absence (−) of 1 nM R1881. Ralaniten induction of expression of metallothionein isoforms was generally not reduced or only modestly impacted by knockdown of AR. Data presented as mean ± SD and analyzed by two-way ANOVA with Sidak’s test applied post hoc (*n* = 3 independent experiments). * *p* < 0.05; ** *p* < 0.01; *** *p* < 0.001; **** *p* < 0.0001; n.s., not significant. ENZ, enzalutamide; RAL, ralaniten; +, R1881; −, EtOH.

**Figure 5 cancers-14-00386-f005:**
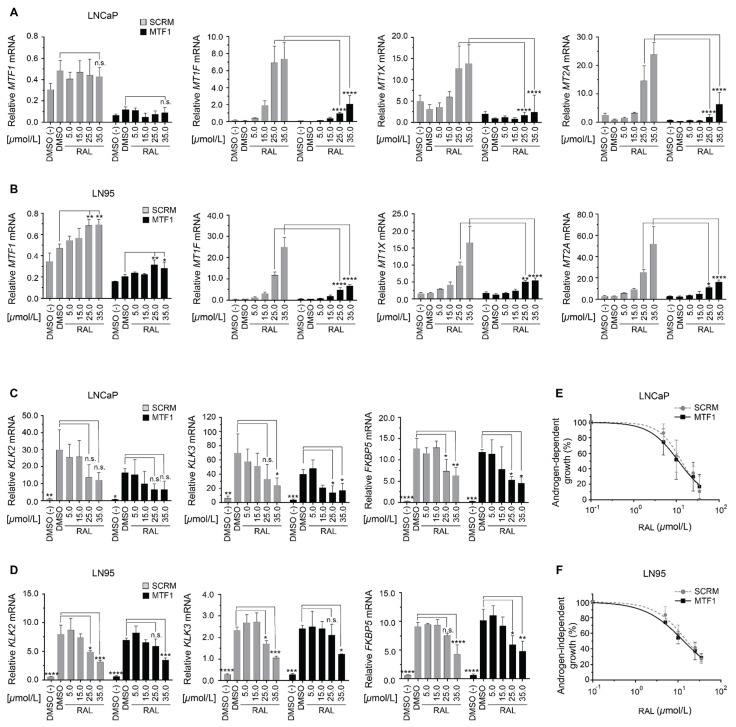
Ralaniten-induced expression of metallothionein is dependent on levels of MTF-1. (**A**) Transcript levels of MTF-1, MT1F, MT1X, and MT2A were normalized to transcript levels of SDHA from LNCaP and (**B**) LN95 cells treated with siRNA targeting MTF-1 followed by incubation with increasing concentrations of ralaniten (5–35 μM), all in the presence of R1881. DMSO is with R1881. Only DMSO (−) was not treated with R1881. (**C**) Transcript levels of KLK2, KLK3, and FKBP5 were normalized to levels of SDHA transcript from LNCaP and (**D**) LN95 cells treated with siRNA targeting MTF-1 with subsequent treatment increasing concentrations of ralaniten (5–35 μM), all in the presence of R1881. DMSO is with R1881. Only DMSO (-) was not treated with R1881. Data presented as mean ± SD and analyzed by two-way ANOVA with Sidak’s test applied post hoc (*n* = 3 independent experiments). (**E**) LNCaP cells were treated with scramble (control) or MTF1 siRNA prior to incubating with 0.1 nM R1881 for 3 days before harvesting and analyzing for proliferation using the crystal violet assay. (**F**) LN95 cells treated with scramble (control) or MTF1 siRNA were incubated in charcoal-stripped serum in the absence of androgen or R1881 for 3 days before harvesting and analyzing cell growth with the crystal violet assay. Data presented as mean ± SD and analyzed by two-way ANOVA with Sidak’s test applied post hoc (*n* = 3 independent experiments). * *p* < 0.05; ** *p* < 0.01; *** *p* < 0.001; **** *p* < 0.0001; n.s., not significant.

## Data Availability

All other relevant data are available from the corresponding author upon reasonable request.

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
