# Peer review of "Differential Gene Expression Profiles between N-Terminal Domain and Ligand-Binding Domain Inhibitors of Androgen Receptor Reveal Ralaniten Induction of Metallothionein by a Mechanism Dependent on MTF1"

_cancers, 2022, doi:10.3390/cancers14020386_

Round 1

Reviewer 1 Report

  Since CRPC is always fatal, and often involves the constitutive activation of the androgen receptor through splice variants that lack the ligand binding domain, compounds that act through this latter site are no longer effective.  Therefore, there is a strong need to develop compounds that target the remaining sites on these truncated proteins.  This manuscript reports the comparisons of the effects of several compounds on the expression of prostate cancer genes, including two EPI compounds that target the N-terminal domain.  It appears to be significant that ralaniten stimulates expression of the metallothionein system and this may be one reason why this compound should no longer be under consideration as a treatment for prostate cancer.  In my mind, the main thrust of this manuscript, however, should be the comparison of ralaniten with EPI 7170.  The data appear to show that this latter compound does not stimulate expression of the metallothionein system, and presumably that is a point in its favor.  The rest of the data are interesting and useful however.  This work is a welcome addition to studies of compounds that attack the NTD and in my opinion, is very important for the future potential of prostate cancer treatments.  

I have one point that lowers my enthusiasm.  The use of the standard error of the mean when only 3 replicates have been performed reduces the error bars by a whopping 41% and leads to an inflated impression on the part of the reader when examining the figures of the variation from replicate to replicate.  It is much more appropriate, for small n values, to use the standard deviation, instead.  I believe that in this case, the error bars would not look so robust as they do with the SEM calculation.

Author Response

R1.1 I have one point that lowers my enthusiasm. The use of the standard error of the mean when only 3 replicates have been performed reduces the error bars by a whopping 41% and leads to an inflated impression on the part of the reader when examining the figures of the variation from replicate to replicate. It is much more appropriate, for small n values, to use the standard deviation, instead. I believe that in this case, the error bars would not look so robust as they do with the SEM calculation.

Authors Response: Figures 2-5, and Figures S1, S3, S5 and associated legends have been amended to include SD rather than SEM as the reviewer has recommended.

Reviewer 2 Report

In the introduction a lot of information about treatment and androgen receptors, but a lack of information about metallothionein.

It is difficult to read the results and the discussion. I suggest switching the methodological part and the results because there was no clarity as to why one or the other cell line was chosen.

The study has been performed in mice also, or has a bioethical authorization been obtained? 

There lack of information about RT-PCR sample replicate. There is no specified real-time version. The new trend for gene expression normalization is normalization done from two or three genes.  For prostate cancer, more suitable housekeeping genes are ACTB, HPRT1, and GUSB. 

Should be mentioned about the treatment is now influenced by gene mutations of BRCA1 / 2 gene mutations also about Ra223.

de novo, in vivo, and in vitro must be written in italic. Also gene names. (line 128, 47, 155,  199, 221).

Should be avoided duplicate information below the figure and text. The description of the figure must go to the same page as the figure.

Text correction: line 155, 175, 211, 216, 281, 333, 419, 420. Hours and hrs in the methods should be adopted the same description. 

Author Response

R.2.1 In the introduction a lot of information about treatment and androgen receptors, but a lack of  information about metallothionein.

Authors Response: We have added the following to the introduction to address the reviewer's concern, and amended the references accordingly.

"Metallothioneins are small (~6 kDa) cysteine rich proteins which are capable of binding numerous essential as well as toxic metal ions, however they preferentially form complexes with zinc[25,26].  Metallothioneins are ubiquitously expressed and play important roles in metal ion homeostasis and detoxification, as well as protecting the cell from oxidative stress[27].  Metal response elements (MREs) have been defined in the proximal promoter elements of metallothionein members of the MT1 and MT2 subfamilies.  Transcription of these genes is markedly enhanced following exposure to zinc and cadmium, primarily through the action of metal regulatory transcription factor 1( MTF-1) binding to the MRE[28]." (lines 84-92)

R2.2 It is difficult to read the results and the discussion. I suggest switching the methodological part and the results because there was no clarity as to why one or the other cell line was chosen.

Authors Response: The reviewer's suggestion on switching the results with the methodological part is unclear, however we agree with the reviewer that the reasons for choosing the particular cell lines was not clear, particularly with respect to the inclusion of LN95 cells.  We hope that the following addresses the concern regarding cell lines.

We initially were searching for differences in the mechanisms of action of AR-NTD inhibitors which we have developed, and existing AR-LBD inhibitors.  Therefore the initial work was performed in LNCaP cells which was a model that was used for much of the preclinical data was generated for ralaniten.  The finding that ralaniten induced metallothionein expression was incidental, and additional cell lines were used to confirm that this was not a phenomenon exclusive to LNCaP cells.  LN95 cells also have a functional AR-FL, however rely upon AR-V7 for growth and do not proliferate in response to androgens unlike LNCaP cells.  As AR-NTD inhibitors target both the full length and AR-splice variants, this was the next logical model to evaluate if these compounds also induced metallothionein expression with a background of constitutively active AR-V7.  DU145 and PC3 cells were used as they lack, or in the case of PC3 express very low levels of non-functional AR.  This helped indicate that the induction of metallothionein genes was independent of the AR and a possible off-target effect by ralaniten.  To better clarify this we have added the following to the text. 

"...on cDNA isolated from LNCaP cells (Figure 1A), a model for which much of the preclinical work in developing ralaniten was performed as well as decades of studies on androgen-regulated genes [16,19,30-32]."  (line 100-102).

"Our next aim was to discover whether the induction of metallothionein expression by ralaniten was restricted to LNCaP cells, or could be replicated in additional models.  To this end, LNCaP95 (LN95) cells were utilized.  Despite expressing functional full-length AR (AR-FL), these cells are reliant upon the splice variant AR-V7 to drive proliferation.  Therefore, the dose-dependence of ralaniten on induction of endogenous expression of metallothionein genes was also measured in LNCaP and LN95 cells. Ralaniten strongly induced expression of metallothionein isoforms in a dose-dependent response in both of these cell lines, demonstrating that the induction of metallothionein expression by ralaniten was not specific to LNCaP cells.  This effect was not measured with any concentration of enzalutamide (Figure 3B)."  (lines 235-244)

"Metallothionein gene expression was assayed in several additional prostate cancer cell lines which lack AR expression (DU145), or express very low levels of non-functional AR (PC3).  Both LNCaP and LN95 cell lines were included as they each had demonstrated increased metallothionein expression in response to ralaniten treatment previously (Figure 3B)." (lines 256-260)

R2.3 The study has been performed in mice also, or has a bioethical authorization been obtained?

Authors Response: The text "All experiments involving animals were conducted in compliance with, and the approval of, the Animal Care Committee of the University of British Columbia (A18-0077)." has been added to line 424-426 under the methods section 3.5.

R2.4 There lack of information about RT-PCR sample replicate. There is no specified real-time version. The new trend for gene expression normalization is normalization done from two or three genes. For prostate cancer, more suitable housekeeping genes are ACTB, HPRT1, and GUSB.

Authors Response: RT-PCR replicate numbers are given in the figure legends in lines 173 and 180 (Figure 2D and 2G), 227 (Figure 3), 310 (Figure 4) and 366 (Figure 5) respectively.  We are unsure what the reviewer is referring to by specified real-time version.  All runs were completed using the QuantStudio 6 Flex Real-Time PCR System as indicated in lines 446-447 under the methods section 3.6.

We routinely use three housekeeping genes in our lab for RT-PCR experiments: ALAS1, RPL13a, and SDHA which have all been previously identified as good housekeeping genes for prostate cancer (Ohl et al, 2005).  For these experiments, ALAS1 and SDHA were used as RPL13a is much more highly expressed (CT~16 for LNCaP and LN95 cell lines) compared to many of the transcripts we were interested in assaying, resulting in very low MNE values.  Conversely both ALAS1 and SDHA have CT values ~24 and ~23 respectively in LNCaP and LN95 cells, much closer to our genes of interest.  Data normalized to SDHA is reported because while both housekeeping genes were quite stable we did find that ALAS1 expression varied slightly across treatments.  Overall trends were highly similar to data normalized to SDHA, however the degree of significance did vary in some instances.      

R.2.5 Should be mentioned about the treatment is now influenced by gene mutations of BRCA1 / 2 gene mutations also about Ra223.

Authors Response: The primary focus of this manuscript was reporting on the discovery of a potential off-target effect of ralaniten, and comparing it to additional AR-NTD antagonists as well AR-LBD antagonists.    Investigating the context of BRCA1/2 mutations was outside the scope of this work, as we were investigating compounds which specifically target the AR.  We do appreciate the reviewer's comment on the importance of BRCA1/2 mutations on the importance of guiding treatment decisions and have added the following to the conclusions section. 

 "Additional evidence is implicating a genomic basis conferring enhanced risk of prostate cancer.  Germline mutations in a number of DNA-repair genes (and especially BRCA2) have been associated with more aggressive disease and poor clinical outcomes [40].  Exploiting alterations in these genes allows their use as additional biomarkers to guide treatment decisions, best exemplified by the recent success of the Phase III clinical trial investigating the use of the PARP inhibitor olaparib compared to conventional AR-targeted therapies (NCT02987543) [41]." (lines 508-514)

R2.6 de novo, in vivo, and in vitro must be written in italic. Also gene names. (line 128, 47, 155, 199, 221).

Authors Response: The text has been amended according to the reviewer's suggestion.

R2.7 Should be avoided duplicate information below the figure and text. The description of the figure must go to the same page as the figure.

Authors Response: We are unsure of what the reviewer is referring to specifically in this regard.  If they could elaborate we would be happy to amend as needed.

R2.8 Text correction: line 155, 175, 211, 216, 281, 333, 419, 420. Hours and hrs in the methods should be adopted the same description.

Authors Response: The text has been amended according to the reviewer's suggestion.  Hrs has been replaced with hours throughout the manuscript.